# The Role of Hyperexcitability in Gliomagenesis

**DOI:** 10.3390/ijms24010749

**Published:** 2023-01-01

**Authors:** Eric A. Goethe, Benjamin Deneen, Jeffrey Noebels, Ganesh Rao

**Affiliations:** 1Department of Neurosurgery, Baylor College of Medicine, Houston, TX 77030, USA; 2Department of Cancer Neuroscience, Baylor College of Medicine, Houston, TX 77030, USA; 3Department of Neurology, Baylor College of Medicine, Houston, TX 77030, USA

**Keywords:** glioma, glioblastoma, tumor microenvironment, hyperexcitability, neuroglial synapse, glutamate, neuroligin-3

## Abstract

Glioblastoma is the most common malignant primary brain tumor. Recent studies have demonstrated that excitatory or activity-dependent signaling—both synaptic and non-synaptic—contribute to the progression of glioblastoma. Glutamatergic receptors may be stimulated via neuron–tumor synapses or release of glutamate by the tumor itself. Ion currents generated by these receptors directly alter the structure of membrane adhesion molecules and cytoskeletal proteins to promote migratory behavior. Additionally, the hyperexcitable milieu surrounding glioma increases the rate at which tumor cells proliferate and drive recurrent disease. Inhibition of excitatory signaling has shown to effectively reduce its pro-migratory and -proliferative effects.

## 1. Introduction

Glioblastoma is the most common malignant primary brain tumor [1,2]. Current treatment consists of maximal safe resection followed by concurrent radiation and temozolomide and adjuvant temozolomide [1]. Despite this, prognosis remains poor, with median survival around 20 months and 5-year survival less than 10% [1,3]. One possible explanation for this is that current treatments are directed at incurring cytotoxicity via DNA damage rather than addressing the heterogenous mechanisms by which glioblastoma progresses [4]. It has recently been established that peritumoral neurons can synapse directly onto glioma cells and that these connections are capable of bona-fide neurotransmission [4]. Furthermore, glioma cells themselves may generate a hyperexcitable microenvironment via non-synaptic glutamate secretion and paracrine activation of glutamatergic receptors [1,5]. Hyperexcitabililty and activity-dependent neurotransmission plays a role not only in tumor-related epileptogenesis but also in tumor cell migration and proliferation, and malignant transformation; concordantly, early EEG evidence of hyperexcitability has been associated with lower overall survival [6,7,8]. This presents several novel targets for therapies directed at curbing the hyperexcitable milieu surrounding glioblastoma, some of which have already undergone human study [4]. In this review we explore the ways in which glioma cells form networks with peritumoral neurons and with themselves and the ways in which glioma interacts with its environment to create pathological levels of excitatory signaling. We also explore role of the hyperexcitable microenvironment in facilitating glioma cell migration, proliferation, and malignant transformation and the ways in which these interactions may be therapeutically targeted. 

## 2. Neuron–Tumor Synapses and Glioma Cell Networks

The discovery of neuron-to- neural or -oligodendrocyte precursor cell (NPC, OPC) synapses, paired with the hypothesis that the putative cell of origin in gliomas is either an NPC or OPC, raised the possibility of neuron–tumor synapses [4]. In support of this, the action of activity-dependent growth signals such as neuroligin-3 (NLGN-3) has been found to cause the expression of synaptic genes in human glioma cells [9]. Thus, recent studies have shown that peri-tumoral neurons synapse directly onto glioma cells and that glioma cells in turn are able to generate postsynaptic currents, spurring the emergence of the field of cancer neuroscience [4,9] (Figure 1). 

Neuron–tumor synapses are vital in the amenability of glioma cells to hyperexcitable signaling and may be identified by electron microscopy as demonstrated by Venkatesh et al. and Venkataramani et al. [9,10]. Work by these authors clearly demonstrates the presence of pre-synaptic neural structures in contact with post-synaptic structures present on glioma cells [9,10]. Venkataramani et al. demonstrate three morphological subtypes of neuron–tumor synapse: (1) synapses directly onto tumor cells, (2) multidirectional synapses onto both tumor cells and other peritumoral neurons, and (3) normal synapses with glioma cells in close, but not synaptic, contact, though the distinct roles of each subtype are not clear [10]. Glioma cells have not been shown to provide presynaptic input to peritumoral neurons [10], but in a recent report by Losada-Perez et al., demonstrated synapses between glioma cells in an in vivo *Drosophila* model [11]. The role of these synapses in gliomagenesis has yet to be elucidated. While NLGN-3 signaling promotes synaptogenesis, inhibition of NLGN-3 was shown to significantly reduce colocalization of pre- and post-synaptic structures in vitro [9].

The synaptic and intercellular networks in which glioma cells are involved create not just a hyperexcitable microenvironment but also an increased sensitivity to excitatory signaling. Consistent with the purported role of hyperexcitability in glioma progression, neuron–tumor synapses are involved in glutamatergic signaling [10]. Venkataramani et al. demonstrated that the electrophysical properties of glioma excitatory postsynaptic currents (EPSCs) were concordant with established values for glutamatergic synapses [10]. These currents demonstrate facilitation in response to paired stimuli and can be inhibited by blockade of excitatory neurotransmission [9]. For example, tetrodotoxin, an inhibitor of voltage-gated sodium channels—and therefore cell depolarization and action potential conduction—was shown to block excitatory post-synaptic currents (EPSCs) in glioma cells [9]. Cyanquixaline and NBQX, both AMPA receptor (AMPAR) inhibitors, similarly blocked glioma EPSCs [9,10]. Taken together these findings demonstrate a clear role of excitatory synaptic communication in the generation of glioma cell electrical activity. These synapses may offer a possible explanation for the seemingly intractable malignant behavior of gliomas, as they are not present on meningioma or oligodendroglioma—two neoplasms that are considered less clinically aggressive [10]. 

The functional connections between glioma cells lend themselves to Increased sensitivity to excitatory signaling. Excitatory synapses between peritumoral neurons and glioma cells are often found on a network of tumor microtubes, and these microtubes connect glioma cells to each other via gap junctions [10]. It has been shown that these cell networks are capable of synchronous electrical activity that is blocked by the application of the gap junction blockers carbenoxolone (CBX) or meclofenolate [9]. These agents reduced the amplitude of intracellular currents, which suggests that the susceptibility of glioma cells to hyperexcitable signaling is in part facilitated by low input resistance due to these junctions [9]. In summary, peritumoral neurons form excitatory synapses with glioma cells, which are connected to each other by gap junctions. AMPAR and gap junction blockers reduce the efficacy of neuron–tumor excitatory signaling.

## 3. The Hyperexcitable Microenvironment

The composition of the extracellular space also contributes to hyperexcitability in glioma. In addition to EPSCs, glioma cells also exhibit slow inward currents (SICs) that are mediated by potassium channels [4]. Prolonged neural activity causes increased extracellular potassium that then causes depolarization of glioma cells; the amplitude of these currents rises in proportion to the degree of neural activity [4,9]. These currents may be elicited with potassium application, even in the presence of pharmacologic blockade of neural activity, and can be blocked by barium, an inhibitor of inwardly rectifying potassium channels [9]. 

Additionally, glioma cells themselves can cause peri-tumoral hyperactivity due to non-synaptic glutamate secretion such that the extracellular glutamate concentration in glioma tissue is 30 times that in peritumoral tissue [1,5]. This secretion is mediated primarily by the cystine-glutamate antiporter, a transmembrane protein that is upregulated in glioma [5,12]. Upregulation of this antiporter, coupled with a lack of functional transporters to clear extracellular glutamate, impairs the reestablishment of glutamate homeostasis [5,12,13,14]. High peritumoral glutamate thus leads to autocrine activation of glutamatergic receptors (AMPA, NMDA, and kainate) [1,5]. The calcium influx caused by activation of these receptors causes further activation of the cystine-glutamate antiporter, leading to higher levels of extracellular glutamate and excitatory neurotransmission [15,16]. Furthermore, AMPARs on glioma cells frequently underexpress mGluR2, an aberration that confers increased calcium permeability, and calcium channels have been found to be upregulated in glioma, which may magnify the effects of these currents [16,17]. Unsurprisingly, expression of the cystine-glutamate antiporter is inversely correlated with survival in GBM patients [18]. Additionally, the cystine imported into glioma cells by the antiporter is converted to cysteine, a rate-limiting precursor for glutathione, an antioxidant that may confer chemo- and radioresistance to glioma cells, making this an attractive target for possible therapies [12]. Furthermore, the secretion of glioma-derived extracellular vesicles has been shown to increase the spontaneous firing of peritumoral neurons, which may not only further activate postsynaptic receptors but is hypothesized by Hua et al. to promote tumor progression via activity-dependent NLGN-3 secretion [13]. 

GABAergic signaling also plays a role in peritumoral hyperexcitability. Peritumoral cortical neurons exhibit downregulated GABAR expression and dysregulated chloride homeostasis, rendering them less susceptible to the increased concentrations of GABA typically found in peritumoral regions [14]. Additionally, peritumoral interneurons are activated by GABA, leading to depolarization of surrounding cortical neurons and further contributing to the hyperexcitable input to tumor cells [14]. The potential contribution of depolarizing GABA currents must be counterbalanced by the early loss of peritumoral interneurons [15], and decreased density of GABAergic synapses in immunocompetent GBM mouse models [6].

The ability of peritumoral neurons to depolarize glioma cells coupled with the resultant increase in peritumoral glutamate can create a feed-forward mechanism in which hyperexcitability begets more hyperexcitability [14]. This may be worsened by the fact that peritumoral cortical networks have been demonstrated to be more sensitive to electrical input and more prone to ictal discharges compared to controls [5]. Chaunsali et al. demonstrated that cortical neurons in xenografted mice generated more frequent action potentials than those of control mice [7]. The pathologically elevated glutamate surrounding glioblastoma, coupled with the increased irritability of the surrounding brain, can predispose patients to seizures [5,14]. Indeed, 30–60% of patients with glioblastoma will develop seizures, and new or recurrent seizures are often a sign of disease progression or recurrence [18]. The tumor microenvironment is characterized by high concentration of potassium and glutamate, which, combined with alterations in receptor function and limited inhibitory signaling, leads to increased neural and tumor electrical activity including the emergence of episodes of cortical spreading depolarization [15]. Ultimately, this neuroglial cross talk is vital to the migration, proliferation, and progression of glioma, and curbing excessive glutamate secretion has shown promise in both seizure and tumor control [14,17].

## 4. Cell Migration and Invasion

Both synaptic and non-synaptic mechanisms contribute to glioblastoma cell migration. Less than 30% of glioblastoma cells receive synaptic input, and the majority of these connections are located in zones of tumor invasion [4]. Venkataramani et al. demonstrated that such glioma cells that are synaptically connected to peri-tumoral neurons and thus able to participate in excitatory neurotransmission have been shown to be significantly more invasive than those that are not connected [10]. These cells may extend beyond the bulk of tumor cells connected by gap junctions; in these cells neuronal activity was shown to increase the speed with which tumor microtubes formed and branched into surrounding tissue, ultimately leading to increased migratory behavior of glioma cells [19]. The inhibition of calcium currents and AMPARs each independently reduced the rate at which tumor microtubes formed and branched, suggestive of a role of excitatory signaling in this facet of cell migration [19].

By simultaneously recording activity-related calcium currents and cell movements, Venkataramani et al. demonstrated that glioma cell migratory episodes corresponded with increases in the frequency of inward calcium currents [10]. The expression of a non-functional AMPAR resulted in reduced migratory behavior [10]. Additionally, the conversion of calcium-permeable AMPAR to calcium impermeable channels via viral transfection was shown to reduce migratory capacity of human glioblastoma cells in vitro [20]. Taken together these findings suggest that excitatory synaptic signaling and the ability of glioma cells to respond to such signals are crucial for cell migration.

Furthermore, autocrine activation of glutamatergic receptors has been demonstrated to lead to increased migratory behavior by modulating adhesion molecule expression and environmental interaction in glioma cells [21]. Piao et al. found that AMPA receptor (AMPAR) stimulation resulted in detachment of glioma cells from the extracellular matrix, an essential step in migration [21]. Additionally, AMPAR activation increased expression of focal adhesion kinases, which in turn facilitate cytoskeletal rearrangement necessary for cell migration [21]. AMPAR expression was also found to correlate with the extent of glioma cell migration in animal models [21]. However, Suina et al. found that NMDA receptor (NMDAR) antagonism, rather than AMPA antagonism, resulted in attenuation of epidermal growth factor (EGF)-driven chemotaxis of glioma cells [22]. Nandakumar et al. demonstrated that NMDAR stimulation led to increased glioma cell migratory behavior in vitro [23]. NMDAR inhibition also resulted in reduced migratory behavior and retraction of extracellular extensions [24]. Thus, enhanced cell migration may lie at the convergence of multiple independent excitatory pathways involving both intracellular and extracellular effectors.

## 5. Proliferation, Tumor Growth, and Recurrence

Both synaptic and non-synaptic excitatory neurotransmission are involved in glioma growth and proliferation. Persistently high intracellular calcium resulting from AMPA or NMDA receptor activation has been shown to promote cell proliferation and tumor angiogenesis [16]. Co-culture of glioma cells alongside neurons has been shown to increase glioma proliferation [9]. This is likely due to synaptic activity, as optogenetic study demonstrates that stimulated glioma cells show increased growth. Venkatesh et al. stimulated blue light channel rhodopsin-expressing glioma mouse xenografts and found significantly increased proliferation compared to controls [9]. 

Overexpression of AMPAR has been shown to increase tumor burden and reduce survival in mouse models of glioblastoma, and non-functional AMPAR led to reduced glioma growth in vivo, but not in vitro, which suggests that synaptic glutamatergic signaling contributes to glioma proliferation [9,10]. Conversely, inhibition of the cystine-glutamate antiporter, which limits the ability of neurons to secrete glutamate and thereby reduces non-synaptic peritumoral hyperexcitability, has been shown to slow glioma growth in mouse models [25]. Furthermore, Ishiuchi et al. transfected glioma cells with a viral vector causing calcium impermeability in AMPARs on glioma cells xenografted into mice [20]. While control cells formed tumors, transfected cells did not form tumors and instead became apoptotic [20]. Transfected glioma cells implanted subcutaneously in mice displayed significantly reduced growth compared to controls and did not respond to AMPAR inhibition, reflective of the role of downstream mediators of excitatory signaling in glioma progression [20]. 

Of note, Oh et al. demonstrated that calcium-permeable AMPARs were selectively upregulated on brain tumor initiating cells compared to differentiated glioblastoma cells [26]. The role of these cells in the invasion, progression, and recurrence further strengthens the role that excitatory neurotransmission plays in glioblastoma.

NMDA receptors may also play a role in glioma proliferation. Ramaswamy et al. demonstrated that NMDAR stimulation significantly increased cell proliferation in vitro in two glioblastoma cell lines [27]. Tsuji et al. found that prolonged stimulation of NMDAR expressed on glioblastoma cell lines in vitro mitigated the cytotoxic effects of temozolomide; conversely, genetic silencing of this receptor significantly potentiated these cytotoxic effects [28]. Additionally, excessive extracellular glutamate can lead to excitotoxicity to peritumoral neurons via NMDAR stimulation, creating a larger niche in which the glioma can grow [29]. 

Patients with postoperative seizures have been shown to have lower time to recurrence than patients who do not, even after adjusting for extent of resection, potentially reflecting a role of neural hyperactivity in recurrence [18]. However, it is unclear whether this effect could be explained by activity-dependent proliferation of glioma cells or simply a more aggressive or infiltrative tumor phenotype, and further investigation is needed to establish causality.

Inhibitory neurotransmission may play a role in glioma proliferation. Glioma cells also express GABA A receptors, though they are often functionally inactive [18]. GABA receptor activation has been shown to slow glioma growth, while inactivation of the receptor by diazepam-binding protein, commonly upregulated in glioma, drives tumor growth [13]. However, the presence of the receptor itself is not necessarily protective, as Wang et al. discovered that the growth inhibiting effects of GABAR-binding microRNA Mir-139-5p are reversed by overexpression of GABRA1 [30]. Ferretti et al. demonstrated the presence of M2 muscarinic receptors, generally considered inhibitory, on human glioblastoma cell lines [31]. Activation of M2 receptors led to reduced cell proliferation in a time- and dose-dependent fashion [31], likely due to impaired cell cycle progression [32].

Other neurotransmitters have also been found to have a role in glioma progression. Neuroligin-3 (NLGN3) is a protein, secreted in an activity-dependent fashion, that has been identified as a mitogen in optogenetic studies of neuron-glioma synapses [33,34]. NLGN3 has been shown to act through the PI3K-Akt-mTOR pathway to induce synaptic gene expression, thus establishing a feed-forward loop in which increased neural activity begets increased tumor sensitivity to this activity [35]. Venkatesh et al. found that tetrodotoxin inhibited NLGN3 release and that NLGN3 knockout resulted in inhibited glioma growth [34]. Mouse models of glioblastoma have demonstrated dose-dependent increases in growth rates with increases in NLGN3 concentrations [36]. Liu et al. demonstrated that the propensity of glioblastomas to recur in the deep structures coincided with a pathologic increase in NLGN3 in these regions, suggesting that this activity-dependent protein can cause more aggressive tumor behavior [36]. Thus, there are multiple ways in which excitatory signaling contribute to malignant behavior in glioma, including AMPA and NMDA activation, limited GABAergic signaling, and neuroligin-3 secretion. These pathways mediate not just cell proliferation but increased resistance to treatment.

## 6. Progression from Low-Grade to High-Grade Glioma

Neuronal activity is important not just for growth and invasion of glioma but also for progression from low- to high-grade tumors. It has previously been demonstrated that gliomas are made of up four general types of cells: OPC-like, NPC-like, mesenchymal-like, and astrocyte-like, with higher proliferation indices seen in OPC-like and NPC-like phenotypes [4]. OPC-like cells have enhanced expression of synaptic genes, suggesting an association between tumor grade and neuronal activity [4]. This association is further strengthened by an observation by Pauletto et al. that patients with low-grade glioma and poor postoperative seizure control had higher rates of malignant transformation than those with good seizure control in univariate analysis, though multivariate analysis failed to confirm this association [37]. As previously mentioned, sulfasalazine has been demonstrated to both delay seizure onset and improve survival, further suggesting a role of hyperexcitability in malignant behavior. Given these discrepant results, further study is needed to clearly define the relationship between hyperexcitability, tumor grade, and survival.

Glutamate signaling plays a role in malignant transformation. De Groot et al. demonstrated that high grade glioma cells have reduced expression of EAAT-2, a transporter responsible for clearing synaptic glutamate, compared to their low-grade counterparts, resulting in increased peritumoral glutamate and suggesting an association between a less-excitable tumor microenvironment and lower grade tumors [38]. Furthermore, restoration of physiologic EAAT2 expression resulted in reduced glioma proliferation in mouse models [38]. Accordingly, a microdialysis study has demonstrated higher concentrations of glutamate in high-grade glioma than in low-grade glioma [39]. Constitutive activation of Akt, a downstream effector of glutamatergic signaling involved with cell viability in glioblastoma, was shown in vivo to convert anaplastic astrocytoma to glioblastoma [40]. Together these data point to a role of excitatory signaling in malignant transformation. NLGN3 expression is associated with decreased survival in patients with glioma, again suggesting a role between activity-dependent signaling and aggressive tumor behavior [33]. Furthermore, expression levels of c-myc, an oncogene that drives metabolism of glutamine to glutamate, are correlated with glioma grade [41]. The malignant transformation of lower-grade tumors may therefore depend on neural activity, pathologic secretion of glutamate via the glutamate-cystine antiporter, the impaired clearance of this glutamate by amino acid transporters, and downstream effectors of glutamatergic signaling. 

## 7. Downstream Pathways

There are multiple downstream mechanisms by which excitatory receptor activation leads to increased malignant behavior. EGFR is involved in several pathways that contribute to cell survival in glioblastoma, including the PI3k/Akt/mTOR and Ras/Raf/MAPK pathways, and can be induced by glutamatergic signaling [42]. The PI3k/Akt/mTOR pathway is involved in protein synthesis, cell growth and proliferation, cytoskeletal reorganization, and, ultimately, cell survival. Continuous activation of this pathway contributes to tumorigenesis and resistance to therapy [43]. The Ras/Raf/mTOR pathway has been shown to contribute to cell growth, differentiation, cytoskeletal reorganization, and membrane trafficking [44]. Schunemann et al. treated glioblastoma cell lines with glutamate and found increased EGFR expression and cell proliferation compared to controls [45]. EGFR expression and cell proliferation were not affected by glutamate when cells were exposed to DNQX, an AMPAR inhibitor, which suggests that EGFR expression downstream from AMPAR activation contributes to glioma progression [45]. This may be due to increased activation of Akt [43]. Interestingly, Akt may be activated by calcium influx via NMDAR and AMPAR and independent of EGFR-related signal transduction [46].

It has been demonstrated in vitro that NMDAR activation leads to decreased levels of gamma-H2AX, a marker of double-stranded DNA breakage, after radiation exposure, which suggests that receptor activation confers radioresistance [24]. This is thought to be caused by signal transduction from NMDARs leading to increased nuclear expression of CREB, a transcription factor that boosts expression of cell survival-related genes [24]. Via CREB expression, NMDAR activation may also induce the expression of cFos, a so-called “early response gene” involved in mitigating radiation-induced oxidative stress [47,48]. NMDAR activation has also been shown to increase expression of MGMT, an enzyme that confers resistance to temozolomide [28].

## 8. Therapeutic Targets

The role of excitatory neurotransmission in promoting glioma progression offers several novel therapeutic targets. These may be directed at postsynaptic effectors, gap junctions, or the mechanisms of synaptogenesis. 

As mentioned above, given its role in maintaining the hyperexcitable microenvironment and conferring treatment resistance, the cystine-glutamate antiporter is a potential therapeutic target. Sulfasalazine, a medication historically used to treat Crohn’s disease, has been shown to inhibit this antiporter and lead to decreased seizure activity in mice, likely due to limitation of peritumoral glutamate secretion [15]. Several authors have investigated sulfasalazine as an adjunct treatment in glioblastoma. In human glioblastoma cell lines, Ignarro et al. found that sulfasalazine administration reduced cell viability both alone and in combination with temozolomide [49]. Chung et al. found reduced rates of tumor growth in glioma-xenografted mice receiving intraperitoneal sulfasalazine injection compared to controls [25]. However, data from human studies are less favorable. An early prospective trial of sulfasalazine in ten patients with recurrent high-grade glioma was terminated early due to a lack of effect and a high rate of toxicity [50]. However, the generalizability of this study is limited, as included patients had a high tumor burden and poor pre-treatment functional status [50]. Takeuchi et al. compared standard chemoradiation with and without the addition of concomitant sulfasalazine in patients with newly diagnosed glioblastoma and found no difference in progression-free or overall survival [51]. However, statistical analysis in this study was complicated by small sample size and a high dropout rate from the sulfasalazine group due to various cytopenias [51]. To our knowledge, no other studies of sulfasalazine in human glioblastoma patients have been published. Given promising results in animal models and the limitations of existing human studies, further investigation with larger study populations is warranted.

Arcella et al. demonstrated reduced growth rates in a mouse model of glioma with LY341945, a glutamate receptor antagonist [29]. MK-801, an NMDAR antagonist, was found to impair both glioma cell migration and proliferation in mouse models [22,23]. Kynurenic acid (KYNA), an inhibitor of both NMDAR and AMPAR, was shown at low doses to inhibit glioma cell migration and at higher doses to inhibit cell proliferation [52]. Memantine, an NMDAR inhibitor, has been shown to reduce glioma cell proliferation in vitro and in vivo, though human studies are ongoing [3]. While these preclinical studies of NMDAR antagonists have demonstrated reduced glioma proliferation, future work with this target will likely be limited by toxicity, as excessive NMDA blockade can lead to psychosis or coma [4]. 

Pharmacologic inhibition of AMPAR has yielded mixed results in phase II trials [14]. Talampanel, a noncompetitive AMPAR inhibitor, was shown to be well-tolerated in patients with epilepsy and thus investigated in patients with glioblastoma. Grossman et al. compared outcomes in patients with newly diagnosed glioblastoma receiving talampanel in addition to standard chemoradiation to historical controls and found improved median survival (20.3 vs. 14.6 months) and two-year survival (41.7 vs. 26.5%) [53]. However, Iwamoto et al. found no effect of talampanel when used as a monotherapy in patients with recurrent high-grade glioma [54]. Venkataramani et al. examined the effects of the AMPA antagonist perampanel in xenografted mice with glioblastoma and found that tumor cell density remained stable over two weeks; tumor cell density doubled over the same period in control mice [10]. While these results are promising, the effect of perampanel on survival in glioblastoma has yet to be investigated in a human study [4]. As activity in AMPA and NMDA receptors has been shown to drive tumor growth, it is likely that the survival benefits conferred by these agents is due to limited excitatory neurotransmission. AMPA receptor blockade may be less limited by side effects in humans. The most common side effects of perampanel are dizziness and somnolence [55], and studies of cognitive effects have demonstrated decrements in attention span and memory speed without affecting global scores of cognition [56].

Despite the limited success of targeting glutamatergic receptors and the association of these receptors with EGFR-dependent pathways involved in cell survival, targeting EGFR has not proven successful in glioblastoma treatment for several reasons [57]. Firstly, the receptor mutations in cancers for which EGFR inhibitors have shown promise differ from those seen in glioblastoma such that EGFR on glioblastoma is not amenable to their effect [57]. Secondly, altered activity in other receptor tyrosine kinases may allow glioblastoma to circumvent the effects of EGFR inhibition [58]. Poor blood–brain barrier penetration has also confounded many trials of EGFR inhibitors [58,59]. However, several trials of newer-generation EGFR inhibitors are ongoing [58].

Postsynaptic calcium currents are a potential therapeutic target. General AMPAR blockers are less effective at inhibiting glioma cell migration and proliferation than blockers of calcium permeable AMPAR [35]. Additionally, calcium influx via voltage-gated calcium channels may be caused by glioma cell depolarization [4]. In a murine model of glioblastoma, Nicoletti et al. demonstrated reduced tumor growth in animals treated with N-type calcium channel inhibitors [60]. Venkataramani et al. point out that pregabalin and gabapentin may represent a possible therapy, as they not only inhibit voltage-gated calcium channels but also have antiepileptic effects and, in the case of gabapentin, may inhibit excitatory synaptogenesis by blocking the action of thrombospondin [4]. Further study is needed to define the role of such agents in glioblastoma treatment.

Given their role in facilitating excitatory signaling, intra-tumoral gap junctions also represent a relevant target. In a mouse model of glioblastoma, DeMeulanaere et al. investigated the gap junction blocker tonabersat and found reduced rates of growth compared to both controls and mice undergoing standard chemoradiation only when tonabersat was added to standard treatment [61].

NLGN3 may represent a therapeutic target, as knockout mice were observed to have otherwise normal neurologic function [34]. Inhibition of ADAM10, the enzyme responsible for NLGN3 cleavage, was shown to reduce glioma growth in mouse models of glioblastoma [34]. Given that NLGN3 signaling is responsible for synaptogenesis, this effect is likely due reduced neuron–tumor synaptic activity.

While hyperexcitability is intimately linked with tumor progression, therapies aimed at limiting presynaptic hyperexcitability have thus far yielded mixed results. In a retrospective study of patients with newly diagnosed glioblastoma receiving standard chemoradiation, Happold et al. found no association between levetiracetam or valproic acid use and survival, though this study had several methodological limitations that curbed its validity and generalizability [62]. However, in a single institution cohort study of isocitrate dehydrogenase wildtype glioblastoma patients, Pallud et al. demonstrated an association between survival and levetiracetam use throughout the entire chemoradiation period compared with its use for part or none of this period (21 vs. 16.8 months) [63].

Therapeutic strategies aimed at curbing excitatory signaling in glioma may address glutamate secretion, glutamate or growth factor receptor activation, calcium currents, gap junctions, or limiting presynaptic activity. While many authors have investigated such opportunities, few have demonstrated promise, reflective of the heterogenous ways in which glioblastoma can evade treatment.

## 9. Conclusions

Both synaptic and non-synaptic mechanisms contribute to excitatory signaling in the invasion and progression of glioblastoma. Glioma cells functionally connect with peritumoral neurons and with themselves, creating a network susceptible to neural activity. Peritumoral hyperexcitability is augmented by antiporter-mediated glutamate secretion, which may contribute to additional excitatory neurotransmission. Activation of glutamatergic receptors on glioma cells leads to increased migratory behavior, proliferation, and malignant transformation of gliomas. Excitatory signaling increases intracellular calcium and leads to extension of tumor microtubes and cellular migration. AMPA and NMDA receptor activation and resultant increases in intracellular calcium also lead to prolonged cell survival, resistance to therapy, and enhanced rates of proliferation, while activation of inhibitory GABA and M2 receptors has been shown to have opposing effects. This creates a system that may be susceptible to becoming a feed-forward loop, where increasing peritumoral neural activity begets more glioma electrical activity and vice versa, thus continuously increasing rates of migration, proliferation, and malignant transformation. Whether such amplification takes place has yet to be established and should be investigated in future study. Genetic and pharmacologic manipulation of this signaling present therapeutic targets, though human studies have yielded mixed results, reflective of the heterogeneity of methods by which glioblastoma progresses and evades treatment.

## 10. Future Directions

Glioblastoma is characterized by extensive molecular heterogeneity and multimodal treatment resistance. Current treatment is aimed at reducing the overall tumor burden and relief of symptoms but does not address the role of hyperexcitable milieu in the development and progression of this disease [4]. The clearly defined roles of excitatory signaling in the progression of glioblastoma as well as early data suggesting that antitumor effects may be achieved with inhibition of this signaling provide impetus for the clinical evaluation of such agents in addition to the current standard of care. However, given the inconsistent results from the limited clinical studies of therapies aimed at curbing hyperexcitable signaling it is likely that effective treatment of glioblastoma lies in simultaneously addressing multiple gliomagenic mechanisms.

## Figures and Tables

**Figure 1 ijms-24-00749-f001:**
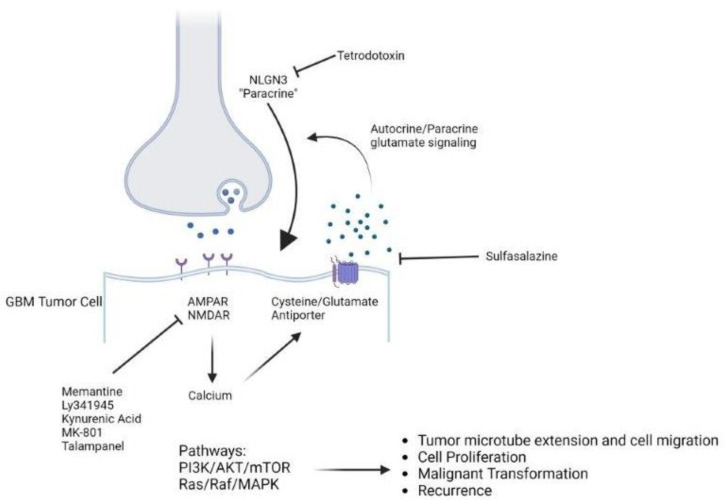
Schematic of a neuron-glioma synapse demonstrating synaptic and non-synaptic avenues of excitatory signaling as well as inhibitors thereof. GBM- glioblastoma; NLGN3- Neuroligin-3.

## Data Availability

Not applicable.

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
