# Peer review of "The Role of Hyperexcitability in Gliomagenesis"

_ijms, 2023, doi:10.3390/ijms24010749_

Round 1
Reviewer 1 Report
Data presented by Goethe et al. in the review 'The Role of Hyperexcitability in Gliomagenesis' explore the role of the hyperexcitable microenvironment in facilitating the migration, proliferation, and malignant transformation of glioma cells and the ways in which these interactions may be therapeutically targeted.
The authors cover the concepts as neuron-tumor synapses and the glioma cell network, the hyperexcitable microenvironment developed by glioblastoma, and the effects of these excitatory signaling on tumor migration, proliferation, growth, and recurrence. Finally, they will address possible therapeutic strategies based on the role of excitatory neurotransmission in promoting the progression of glioma.
In general, it seems to me a very new and interesting subject; proof of this are the references of the last years cited, therefore I would like to congratulate the authors. Despite this, below I mention some aspects that could help improve the text:
2. Neuron-Tumor Synapses and Glioma Cell Networks
In this section, the authors address the synaptic and non-synaptic relationship between glioma and peritumoral cells.
Line 53. When describing the morphological subtypes of the neuron-tumor synapse, the third type is a little confusing: "normal synapses with glioma cells in close, but not synaptic, contact" Are the authors referring to autocrine/paracrine glutamate signaling?
Line 55. They then mention that "Glioma cells have not been shown to form synapses with other glioma cells, nor have they been shown to provide presynaptic input to peritumoral neurons”, but recently Losada-Perez et al.2022 (PLoS Genet 18(7): e1010329) describe, in Drosophila, synapses between GB cells (intratumoral synapases) and suggest the possible role of these in tumor progression. Could the authors discuss this point?
Line 66. The authors describe excitatory postsynaptic currents (EPSCs) of glioma and facilitation (increase in EPSCs slope) in response to paired stimuli (consecutive action potentials). What is the function of these EPSCs in glioblastoma cells? Are action potentials generated in glioma? Have voltage-gated channels been described in glioma? It would be interesting to discuss these points in the text.
3. The Hyperexcitable Microenvironment
In this section the authors discuss hyperexcitable microenvironment induce by glioma, through different mechanism as: Slow inward currents (SICs) that are mediated by potassium channels in glioma cells (increased extracellular potassium). The up-regulated cystine-glutamate antiporter and lack of functional transporters to clear extracellular glutamate lead to autocrine activation of glutamatergic receptors. Glioma AMPA receptor with increased calcium permeability. Peritumoral neurons exhibit down-regulated GABAR expression and dysregulated chloride homeostasis… All these changes make patients more susceptible to seizures.
In this section, I have the following questions: Are these changes in the entire tumor or only on its surface? Do these changes occur over the entire surface or only in specific regions? For example, in the areas where migration is being activated
4. Cell Migration and Invasion
In this section, the authors describe the role of synaptic and non-synaptic mechanisms contribute to glioblastoma cell migration.
Line 135. There is a repetition 'non-synaptic'
In this section, I would like to clarify the relationship between receptors, microtubules, and cell migration. Maybe a figure could help this?
5. Proliferation, Tumor Growth, and Recurrence
This section describes the effects of synaptic and non-synaptic excitatory neurotransmission on the growth and proliferation of glioma. In addition, they discuss the effects of inhibiting the different receptors and proteins involved in this hyperexcitability on proliferation, growth, and tumor recurrence.
Even though part of these effects are reflected in Figure 1, perhaps the authors could elaborate a more detailed figure where all the commented aspects are reflected.
6. Progression from Low-Grade to High-Grade Glioma
Next, they discuss the role of glutamatergic synapsis in the malignant transformation of the tumor.
Line 236. Explain in a clearer way the observations of Pauletto et al.
7. Downstream Pathways
This is a very interesting section, in which the authors discuss the possible downstream mechanisms by which excitatory receptor activation leads to increased malignant behavior. They raise mediators such as: EGFR, AKT or CREB, activated by AMPA or NMDA receptors in glioma cells.
This is one of the keys to the effects mediated by hyperexcitability. It would be interesting if this section were expanded. Although I understand that these pathways are probably not known, the authors would dare to hypothesize what is happening.
Perhaps a summary scheme could improve this section
8. Therapeutic Targets
This is a very nice and interesting section, in which the authors explain how the excitatory neurotransmission describing process in glioma offers several novel therapeutic targets to attack glioma progression.
In this section there are clear differences in the results of animal models (mice) and human. The no effects in human experiment are attributed to a low number of subjects. But could these differences be due to differences between the two species? Perhaps the mechanisms activated between both species are different? Could the authors discuss this aspect in the text, please.
Could the authors also discuss the side effects associated with the inhibition of AMPA or NMDA receptors, in aspects such as memory or other aspects.
9. Conclusion and 10. Future Directions
In the conclusions, perhaps I would try to highlight the relationship between the activation of glutamatergic receptors and how this activation increases increased migratory behavior, proliferation, and malignant transformation of gliomas.
GENERAL COMMENTS
The work would improve by putting a summary sentence at the end of each section, which concludes what was discussed in the said section and links to the next section.
The work would be much better with more illustrations. Figure 1 is fine, however, I would propose a more complete figure that reflects the different aspects dealt with, or different specific figures of each section.
Extracellular vesicles (EVs) play an indispensable role in neuron-glioma interaction in the brain microenvironment. The authors believe that EV-glioma could be involved in the hyperexcitability induced by glioma cells.
Recently, a review by Hu et al. 2022, “Glioma‑neuronal interactions in tumor progression: mechanism, therapeutic strategies and perspectives”, describes the roles of neurotransmitters in neuron‑glioma interactions. What do the authors think about this review? It addresses many aspects like yours, although it does not specifically focus on hyperexcitability. What do the authors think makes their work different from this one?
In the present review, it has not yet been established whether hyperexcitability increased migratory behavior, proliferation, and malignant transformation of gliomas. They describe that this hyperexcitability is necessary for gliomas changes, but how does this mediate this? Maybe the authors could hypothesize at the end of the review.
Reviewer 2 Report
The manuscript presented by Eric Goethe and his colleagues is a comprehensive overview of discussing the role of hyperexcitability and glioma genesis. Overall, I found this paper is comprehensive and well-balanced review of the literature. I only have a few minor suggestions.
In the section "Downstream pathways", the author discussed a lot in a relatively short section. In order to help reader to understand the related concepts better, I would recommend to add a simplified scheme to show these related pathways directly.
Also, they are some typos in this paper, please fix them.
Author Response
The manuscript presented by Eric Goethe and his colleagues is a comprehensive overview of discussing the role of hyperexcitability and glioma genesis. Overall, I found this paper is comprehensive and well-balanced review of the literature. I only have a few minor suggestions.
In the section "Downstream pathways", the author discussed a lot in a relatively short section. In order to help reader to understand the related concepts better, I would recommend to add a simplified scheme to show these related pathways directly.
Author response: Thank you for your feedback. We have added these pathways to figure 1 in order to clarify their relationship with the rest of the content discussed in this paper.
Changes to text: Please see revised Figure 1.
Also, they are some typos in this paper, please fix them.
Author response: Thank you for your feedback. We have proofread the paper and made appropriate changes.